# Sensing the *ortho* Positions in C_6_Cl_6_ and C_6_H_4_Cl_2_ from Cl_2_^−^ Formation upon Molecular Reduction

**DOI:** 10.3390/molecules27154820

**Published:** 2022-07-27

**Authors:** Sarvesh Kumar, José Romero, Michael Probst, Thana Maihom, Gustavo García, Paulo Limão-Vieira

**Affiliations:** 1Atomic and Molecular Collisions Laboratory, CEFITEC, Department of Physics, Campus de Caparica, Universidade NOVA de Lisboa, 2829-516 Caparica, Portugal; s.kumar@campus.fct.unl.pt (S.K.); j.romero@campus.fct.unl.pt (J.R.); 2Institut für Ionenphysik und Angewandte Physik, Leopold-Franzens Universität Innsbruck, Technikerstraße 25, 6020 Innsbruck, Austria; 3School of Molecular Science and Engineering, Vidyasirimedhi Institute of Science and Technology, Rayong 21210, Thailand; 4Department of Chemistry, Faculty of Liberal Arts and Science, Kamphaeng Saen Campus, Kasetsart University, Nakhon Pathom 73140, Thailand; t_maihom@hotmail.com; 5Instituto de Física Fundamental, Consejo Superior de Investigaciones Científicas, Serrano 113-bis, 28006 Madrid, Spain; g.garcia@csic.es

**Keywords:** hexachlorobenzene, dichlorobenzene, charge transfer, collision induced dissociation, geometric effect

## Abstract

The geometrical effect of chlorine atom positions in polyatomic molecules after capturing a low-energy electron is shown to be a prevalent mechanism yielding Cl_2_^−^. In this work, we investigated hexachlorobenzene reduction in electron transfer experiments to determine the role of chlorine atom positions around the aromatic ring, and compared our results with those using *ortho*-, *meta*- and *para*-dichlorobenzene molecules. This was achieved by combining gas-phase experiments to determine the reaction threshold by means of mass spectrometry together with quantum chemical calculations. We also observed that Cl_2_^−^ formation can only occur in 1,2-C_6_H_4_Cl_2_, where the two closest C–Cl bonds are cleaved while the chlorine atoms are brought together within the ring framework due to excess energy dissipation. These results show that a strong coupling between electronic and C–Cl bending motion is responsible for a positional isomeric effect, where molecular recognition is a determining factor in chlorine anion formation.

## 1. Introduction

Hexachlorobenzene (C_6_Cl_6_) has caught the attention of the international scientific community due to its role as an atmospheric pollutant which is frequently detected across the globe [1]. It is removed from the troposphere by photolysis and deposition to soil and water [2]; its net residence is about one year. Hexachlorobenzene enters the environment as a byproduct from the manufacture and use of chlorinated solvents, the application of C_6_Cl_6_-contaminated pesticides and the incineration of C_6_Cl_6_-containing wastes. Long-range transport from its origin source [3,4] also contributes to the spread C_6_Cl_6_ in the environment. In the atmosphere, C_6_Cl_6_ can undergo reactions with ^•^OH radicals, although this sink mechanism is not particularly efficient due to a relatively low-rate constant at room temperature, i.e., 0.27 × 10^−13^ cm^3^ molecule^−1^ s^−1^ [3]. Ormad et al. [4] and Roche et al. [5] showed that hexachlorobenzene can be eliminated by ozone coupled with hydrogen peroxide processes, and these can be enhanced by combination with active carbon absorption methods. Additionally, at the local level, the removal of C_6_Cl_6_ can be performed through incineration processes at high temperatures, with a conversion efficiency of 70% at ~1300 K [6]. Nonetheless such reactions yield elemental Cl_2_ and polychlorinated aromatic compounds (e.g., C_2_Cl_6_, C_5_Cl_6_), which can be as toxic as hexachlorobenzene [2,6]. Given the low solubility of chlorobenzenes and their tendency to vaporize, other relevant processes to chemically change such compounds have been reported. Amongst them, anaerobic metabolism under methanogenic conditions (metabolic byproduct in hypoxic conditions) can be cited [7,8]. These metabolic processes are mainly based on the dichlorination of hexachlorobenzene into tri- and di- chlorobenzenes in anaerobic municipal sewage sludge conditions. Additionally, using anaerobic biotransformation, Bosma and co-workers [9] reported the reductive dichlorination of trichloro- and dichlorobenzene isomers. However, as shown in previous studies [10,11,12,13,14,15,16], one of the most effective ways to chemically dissociate complex molecules is via electron attachment, i.e., either directly irradiating the target with free low-energy electrons or using electron transfer mechanisms. We have also shown that the latter can be bond-breaking selective, thereby offering the possibility of controlling the outcome channels of the reaction [17].

Until recently, the electronic state spectroscopy of hexachlorobenzene had not been investigated comprehensively [18,19]. Nonetheless, a few studies on vibronic interactions and single charge transfer, and theoretical and experimental methodologies to obtain molecular electron affinities have been reported [20,21,22,23]. Electron attachment and/or charge transfer techniques are excellent tools to probe the role of different electronic states of a molecule. The latter is particularly relevant in terms of assessing molecular states with positive electron affinities. Upon electron capture, the extra charge is temporarily accommodated in the field of the molecule, resulting in a short-lived anion that may be stabilized by the internal redistribution of the excess energy and/or dissociate, yielding a neutral radical and a fragment anion. Such resonances are pertinent in a wide range of chemical reactions of molecular compounds which are prevalent in, e.g., radiation-induced biological damage [24], the interstellar medium [25] and atmospheric chemistry [26]. Regarding negative ion formation, recently, Kumar et al. [27] investigated charge transfer processes in potassium-neutral hexachlorobenzene collisions, yielding seven different channels of negative ion formation. In the wide lab frame collision energies probed (10–100 eV), the formation of a non-dissociated parent anion, C_6_Cl_6_^−^, as well as other fragment ions stemming from the loss of neutral chlorine atoms and complex internal reactions within the temporary negative ion, was reported [27].

In the present joint experimental and theoretical study, we were particularly interested in investigating the underlying molecular mechanisms responsible for Cl_2_^−^ formation upon electron transfer from a neutral potassium atom to hexachlorobenzene and dichlorobenzene neutral molecular targets. In particular, we comprehensively show the internal routes within the temporary negative ions formed from such chlorinated molecules which are responsible for selective C–Cl bond excision. The geometric effect of the chlorine atom positions around the aromatic rings and the presence of a barrier result in an energy balance that favors *ortho* configurations.

## 2. Results

Electron transfer from a potassium atom (K) to a polyatomic molecule (ABC) can be summarized by Equation (1):K + ABC → K^+^ + ABC^•−^ → K^+^ + AB^−^ + C^•^(1)
where K^+^ denotes the oxidized projectile and ABC^•−^ the reduced molecular ion, also known as the transient negative ion (TNI). The TNI can have a long lifetime relative to a competing autodetachment process, allowing it to be detected within the experimental μs time window regime. In such a redox reaction, the energy deposited in the TNI may be redistributed through the different internal degrees of freedom, while direct and statistical dissociation may lead to bond excision, usually with the excess electron at a highly electronegative site of the molecule, e.g., halogen atoms [28] or the nitro group in radiosensitizers [24,29,30].

This is the first combined electron transfer investigation of hexachlorobenzene (C_6_Cl_6_) and dichlorobenzene isomers (*o*-, *m*-, *p*-C_6_H_4_Cl_2_) yielding a fragment anion at m/z 70, corresponding to the loss of 1,2,3,4-tetrachloro-1,3-cyclohexadien-5-yne (Equation (2)) and benzyne (Equation (3)) radicals from the TNIs:K + C_6_Cl_6_ → K^+^ + C_6_Cl_6_^•−^ → K^+^ + Cl_2_^−^ + C_6_Cl_4_^•^(2)
K + C_6_H_4_Cl_2_ → K^+^ + C_6_H_4_Cl_2_^•−^ → K^+^ + Cl_2_^−^ + C_6_H_4_^•^(3)

The TNIs are formed by accommodating the extra electron into the π* antibonding molecular orbitals of the neutral molecules, while internal rearrangement and efficient intramolecular electron transfer via diabatic curve crossing with σ*(C–Cl) may yield a radical and a negative ion, as shown in Equations (1) and (2). These experiments were performed in a crossed molecular beam setup with the key relevant reaction mechanism calculated by quantum chemical methods (see Section 4 for experimental and theoretical details).

We show that the collision-induced dissociation mechanism is strongly dependent on the geometrical position of each Cl atom participating in Cl_2_^−^ formation, making it possible to control the specific bonds to be cleaved upon electron transfer, i.e., selective molecular bond excisions just in the *ortho* positions are favored by the capture of an extra electron coupled with relevant internal degrees of freedom along the C–Cl bond.

## 3. Discussion

Figure 1 shows the time-of-flight (TOF) mass spectrum with the *m*/*z* most intense anions forming in the electron transfer from K to C_6_Cl_6_ at 55 eV lab frame energy (43.5 eV in the center-of-mass frame); a discernibly less intense anion at *m*/*z* 70 also formed. Such an anion formation involves a concerted intramolecular rearrangement mechanism within the TNI, where two C–Cl bonds have to be cleaved and a Cl_2_ molecule formed with the extra charge. The detection of the parent anion in the present experiments means that hexachlorobenzene has a positive electron affinity (EA), i.e., the anionic state lies energetically below the neutral state. The present quantum chemical calculations at the CBS-QB3 level of theory predict a value of +1.022 eV, thus supporting the experimental observation. This is also in good agreement with results reported by Kumar et al., i.e., +1.07 eV from the *ω*B97XD/aug-cc-pVDZ level [31].

The branching ratios (BR) shown in Figure 2 indicate that the chlorine molecular anion formation was restricted to less than 10% of the total anions recorded in a wide range of collision energies. Nonetheless, above 45 eV, it gained some intensity at the expense of the decreasing tendency of C_6_Cl_6_^−^. The rather constant behavior above 55 eV is reminiscent of the fast collision dynamics (*t*_collision_ < 40 fs), where electron promotion into strongly antibonding σ*(C–Cl) potential energy surfaces above the ground state is prevalent.

The computational results for the thresholds of Cl_2_^−^ dissociation are shown in the Appendix A (computational details in Section 4.2 below) for hexachlorobenzene and di-chlorobenzene. Using the bond dissociation energies of C_6_Cl_5_–Cl, C_6_Cl_4_–Cl and Cl–Cl (1.97, 2.39 and 3.66 eV for *o*-, *m*- and *p*-C_6_Cl_6_, respectively), the reaction threshold was obtained at 5.88 ± 0.30 eV from the K^+^ energy loss experimental data of Kumar et al. [27]. If we subtract the potassium ionization energy (4.34 eV), the threshold is now expected to be 1.54 ± 0.30 eV. Thus, we assigned Cl_2_^−^ formation to the exclusive cleavage of two close-lying C–Cl bonds from C_6_Cl_6_ upon electron transfer; the experimental value was in good accord with the quantum chemical calculations for the *o*-C_6_Cl_6_ configuration.

Electron transfer from potassium collisions can create anions that will be initially in an electronically excited state. Due to the energy available in the center-of-mass system, several molecular orbitals are accessible, even those above the dissociation limit which can serve as resonances. This primary process is difficult to disentangle, but all excess energy above the dissociation threshold will be converted into kinetic energy. We studied the system moving inside its ground state potential energy hypersurface by investigating the possible reaction pathways. Notably, the kinetic features of the Cl_2_^−^ (and Cl_2_) detachment reaction for *ortho* and *para*-Cl migration were studied by calculating the shortest reaction pathways for Cl migration (Figure 3 and Figure 4, respectively).

In the *ortho* case (Figure 3), at the transition state TS1, one Cl–C bond was weakened while Cl formed a bond with the adjacent C. At TS2, both C–Cl bonds were weakened and a Cl–Cl bond was formed. While all transition states are naturally exothermic, the anionic system requires much less energy to overcome the barriers. An analogous pathway was derived for the detachment reaction involving two Cl atoms at para positions (Figure 4). As shown, a large number of intermediate steps was necessary, rendering this reaction unlikely (details of the calculations of the reaction pathways are given in Section 4.2). It should be mentioned that the relative product energies given in Figure 3 and Figure 4 (for the ortho case +2.41 eV for detachment of Cl_2_^−^ and 4.31 eV for detachment of Cl_2_) were indeed close to the threshold energies listed in Appendix A. Since the M06-2X functional and the basis set used are geared toward accurate and transition state energies, they were expected to be somewhat less accurate for the thresholds than the values in Appendix A.

In electron transfer to hexachlorobenzene, the minimum energy required to break a second C–Cl bond (*o*-C_6_Cl_4_-Cl, *m*-C_6_Cl_4_-Cl and *p*-C_6_Cl_4_-Cl) lies between 2.7 and 4.4 eV (Table 1), so site selectivity yielding Cl_2_^−^ does not result from any particular energy constraint.

The electronic structure of the associated TNIs accessed in the collision process must play a significant role in the intramolecular mechanisms which are responsible for channeling the excess energy through the different available degrees of freedom that will lead to molecular dissociation. In order to investigate this, we conducted a comprehensive dynamic investigation of the role of C–Cl stretching in the TNI. Figure 5 shows the changes in energy (ΔE), Cl–Cl distance and the shape of the C_6_Cl_6_^−^ singly occupied molecular orbital (SOMO). The C–Cl bond stretching distances of two Cl atoms in the *ortho* isomer are varied while keeping all other geometric parameters relaxed. The color maps show the electron density of the SOMO, squared and integrated along the z-coordinate perpendicular to the molecular plane. A close inspection of this figure shows that the equilibrium geometry is found at 1.802 Å, while at P1, the C–Cl distance is considerably shorter. At P2, the SOMO is delocalized over the whole anion, whereas from P3 to P6, the C–Cl bond elongation causes a SOMO displacement to two neighboring C–Cl bonds. The energy increased, although at 2.4 Å, there was no evidence of bond breaking, i.e., Cl atoms did not move together and there was no indication of a bond having formed between the two adjacent chlorine atoms.

The other possible mechanism is related to the change in the Cl–Cl distance of two adjacent Cl atoms as the C–Cl bending distances changed while relaxing the other geometric parameters within the TNI (Figure 6). The Cl–Cl distance of the two upper Cl atoms increased from P1 to P6. The equilibrium geometry was found at 3.246 Å, while from P1 to P5, the SOMO was delocalized and did not appreciably change upon increasing the Cl–Cl distance. In P6, the occupied MO moved to the elongated C–Cl bonds and their Cl atoms. The Cl–Cl distance increased further, in line with the observation from the scan of the C–Cl coordinate, whereas the C–C–Cl angles remained at 120°. However, as the Cl–Cl distance decreased, at ~2 Å, a bond formed between two adjacent chlorine atoms, and these detached from the neutral ring carrying the extra charge, as confirmed from the atomic charge populations.

The hypothetical symmetric direct abstraction of Cl_2_^−^ was also studied in another way. If the Cl-Cl distance in the remaining Cl_2_ was clamped at the equilibrium value of the free diatomic, we observed a barrier in the neutral system but none in the anion (Appendix A). This indicated that such a one-step detachment of Cl_2_ is not completely impossible but would require an excitation of the anion that forces two *ortho* Cl atoms into the appropriate distance. Details of the calculations are given in the Appendix A, Section 4.

The electron transfer TOF mass spectra of negative ions formed in potassium collisions with dichlorobenzene isomers (Equation (3)) did not show a parent anion formation, but were dominated by Cl^−^, with Cl_2_^−^ only being formed from 1,2-C_6_H_4_Cl_2_ (see SM). The absence of a non-dissociated anion was not surprising, given the negative EA values (at the CBS-QB3 level of theory) of −0.20, −0.21 and −0.44 eV for *o*-C_6_H_4_Cl_2_, *m*-C_6_H_4_Cl_2_ and *p*-C_6_H_4_Cl_2_, respectively, meaning that no thermodynamically stable anions were present.

Another interesting aspect of the dichlorobenzene isomer fragmentation pattern pertains to the absence of a C_6_H_4_^−^ ion despite the benzyne radical electron affinity (EA = 1.265 ± 0.008 eV [32]). Upon reduction of C_6_H_4_Cl_2_ molecules in a low-energy collision regime (typically below 100 eV), the extra charge was accommodated in the π* antibonding MOs of the ring. As long as an efficient diabatic curve crossing enabled a fast-intramolecular electron transfer to a σ*(C–Cl) bond, the collision induced dissociation favored the formation of fragments with higher electron affinities (EA(Cl) = 3.6131 eV [33], EA(Cl_2_) = 2.50 eV (Table 1)), viz. Cl^−^ and Cl_2_^−^. Note that the formation efficiency of the latter relative to the former anion was lower because two C–Cl bonds had to be excised and a molecular ion formed. This certainly required a concerted mechanism within the TNI and energy redistribution through the available degrees of freedom.

The CBS-QB3 (and G4MP2) level of theory calculations in Table 2, show the Cl_2_^−^ thermodynamic thresholds for *o*-, *m*- and *p*-C_6_H_4_Cl_2_, to be 2.08, 2.78 and 3.29 eV, respectively, by taking the bond dissociation energies of C_6_H_4_Cl–Cl, C_6_H_4_–Cl, Cl–Cl (Table 1) and the Cl_2_ electron affinity (Table 1). Thus, the proximity of two chlorine atoms in the ring, as was the *o*-C_6_H_4_Cl_2_ case, required the least energy for Cl_2_^−^ formation among the three dichlorobenzene isomers, i.e., ~2.1 eV. In *m*-C_6_H_4_Cl_2_ and *p*-C_6_H_4_Cl_2_, the reaction energies exceeded 2.8 eV; however, Cl_2_^−^ formation from these molecules was not observed in the experiment. As happens in *o*-C_6_Cl_6_, the geometrical proximity of Cl atoms is pivotal to guarantee the formation of the Cl_2_^−^ anion. Figure 4 indicates that for Cl atoms in the *para*-positions, the reaction would require too many steps to be viable.

Comparing the negative ion formation from C_6_Cl_6_ and C_6_H_4_Cl_2_ molecules, the remarkable *ortho* position sensitization in the presence of an electron donor triggered a specific electron transfer process with the same consecutive C–Cl bond excision, leading to the formation of a neutral radical and the corresponding molecular fragment anion. The calculated energy thresholds indicated no appreciable difference in the exclusive reaction, yielding Cl_2_^−^ from *o*-C_6_Cl_6_ and *o*-C_6_H_4_Cl_2_, whereas in the case of the former molecule, this was evident by the experimental value obtained. The preferred selectivity of the bond excision position in these molecules was not particularly enhanced because the electron transfer reaction proceeded in the presence of barriers (transition states), leading to a comparably low abundance of the fragment chlorine molecular anion formed; see SM. This notwithstanding, the process investigated here may be of relevance to key selected molecular compounds playing a specific role in catalysis, enzymatic reactions or, more generally, in chemical reactions where a transition state is prevalent in bond breaking and making [34].

## 4. Materials and Methods

### 4.1. Experimental Setup

A crossed atom-molecular beam setup at the Universidade NOVA de Lisboa, Portugal, was used to obtain the experimental data. The experimental methodology used in this study has been extensively explained elsewhere [17,35]; therefore, just a brief description will be given here. It comprised two vacuum chambers, both differentially pumped and interconnected by a gate valve with a 0.5-cm wide aperture. The base pressure in the potassium chamber was 4 × 10^−5^ Pa, while in the collision chamber, it was 5 × 10^−5^ Pa. The working pressure in the collision chamber after sample effusion was typically 1 × 10^−3^ Pa. In the potassium chamber, a commercial potassium ion source (HeatWave Labs, Watsonville, CA, USA) generated hyperthermal potassium cations (K^+^_hyp_) that were accelerated to a set level of kinetic energy toward the entrance of an oven, where they engaged in a resonant charge exchange (CEO) with thermal potassium atoms (K^o^_th_), obtained by heating solid potassium at 393 K, yielding K^o^_hyp_. The resultant beam comprised K^+^_hyp_ ions that did not charge exchange and were removed from the K^o^_hyp_ beam by two deflecting plates placed at the exit of the CEO, before passing into the collision region. From the resonant charge-exchange process and the CEO slits apertures, the K^o^_hyp_ beam was mainly composed of potassium atoms in the ground state configuration with its outermost electron as 4s. Thus, the experimental thresholds of formation assumed that K* in a 4p state would result in values at lower energies than those reported in these experiments. Such behavior has been previously reported in other energy loss data from potassium collisions with pyrimidine [36], halothane [37], tetrachloromethane [28] and methanol [38]. The K^o^_hyp_ beam intensity was monitored at the entrance of the collision chamber by a surface ionization detector of the Langmuir-Taylor type. Subsequently, the K^o^_hyp_ beam crossed at a right angle with an effusive target beam, which entered a vacuum through a 1-mm diameter capillary from an external sample holder. The negative ions formed by electron transfer in the collision of neutral potassium atoms with the target neutral molecules were extracted by a pulsed electrostatic field (340 Vcm^−1^) and mass analyzed by a linear TOF spectrometer with an estimated mass resolution of 125. The beam energy resolution for TOF mass spectra collection in the collision energy range investigated was ~0.6 eV. Mass calibration was performed using the well-known fragmentation patterns from collisions of potassium atoms with nitromethane [39] and/or carbon tetrachloride molecules [28]. Background measurements were obtained and subtracted from the sample measurements.

The energy loss of potassium cations formed after the collision experiments was analyzed in the forward scattering direction, i.e., such experiments were not performed in coincidence with TOF mass spectrometry. The analyzer was operated in constant transmission mode, thereby keeping the resolution constant throughout the entire scan. The estimated energy resolution during the experiments was ~1.2 ± 0.2 eV. The energy loss scale was calibrated using the K^+^ beam profile from the potassium ion source serving as the elastic peak. Hexachlorobenzene (C_6_Cl_6_) and 1,2-, 1,3- and 1,4-dichlorobenzene (C_6_H_4_Cl_2_) were supplied by Aldrich with a stated purities of ≥98%, 99%, ≥99% and ≥99%, respectively. The liquid samples (1,2- and 1,3-C_6_H_4_Cl_2_) were degassed through repeated freeze-pump-thaw cycles. The solid samples (C_6_Cl_6_ and 1,4-C_6_H_4_Cl_2_) were used as obtained and gently heated to 340 K and 363 K through a temperature PID (proportional-integral-derivate controller) unit. In order to test for any thermal decomposition products within the different molecular beams, mass spectra were recorded at different temperatures and no differences in the relative peak intensities as a function of temperature were observed.

### 4.2. Computational Methods

A variety of quantum chemical methods was employed to obtain information about the dissociation reactions.

(a)Energy thresholds for the dissociation of Cl, Cl_2_ and their anions from C_6_Cl_6_ and C_6_H_4_Cl_2_ were calculated by means of the quantum thermochemical extrapolation methods G4MP2 [40,41] and CBS-QB3 [42,43]. These semi-empirical compound schemes are known to accurately predict the energy differences between reactants and products. For comparison, straightforward density functional calculations with the B3LYP-GD3 functional [44] with dispersion corrections [45] and the aug-cc-pVTZ basis-set [46] were also performed and were in accordance with the expected tolerances. (Appendix A). Since the three model chemistries were based on independent assumptions, this lent credibility to the reported values.(b)Energy thresholds contain information about reaction thermodynamics but not about kinetics. Therefore, we covered possible reaction pathways by means of transition state searches using the Berny algorithm [47] to move from reactants to products. Specifically, a lowest-energy pathway where one Cl atom moved toward another one until Cl_2_ abstraction was possible was calculated. The transition states along the pathways had to have one vibrational mode corresponding to the shift from reactants to intermediates (of reactants) with an imaginary frequency, while all other vibrational modes indicated the local energy minima of their normal coordinates. The intermediates were isomers with all vibrational frequencies being real numbers. We verified that these conditions had been fulfilled. The pathway calculations were performed with the M062X functional [48] and the 6-31G (d,p) basis set [49,50]. Transition state energies are not well covered with many standard functionals, and M062X was specifically developed to obtain accurate barriers. This relatively small basis set is often used together with the M062X functional, since it allows for the large number of single-point calculations and optimizations imposed by the search procedure. No geometrical restraints were imposed in the reaction pathway calculations.(c)Vibrational frequencies were also calculated at the B3LYP-GD3/aug-cc-pVTZ level, as were the various orbital energies and charge distributions (Appendix A). The same functional and basis set were used in the pointwise relaxed scan of the potential energy curve for the symmetric detachment of Cl_2_^−^. (Figure 5, lower panel and Appendix A in the Appendix A). This functional augmented with corrections for exchange contributions, together with a reasonably large basis set including diffuse functions, is well established for energies and structures, including those of anionic systems.(d)B3LYP-GD3/aug-cc-pVTZ method and basis set were also used in studying the symmetric direct abstraction of Cl_2_^−^. (Appendix A Section 4 and Figure 6). Calculations of the anion started from trial wavefunctions, where the Cl_2_ fragment already carried a negative charge and the C_6_Cl_4_ fragment was neutral. In this way, we circumvented the use of multiconfigurational methods that would otherwise be necessary to describe smooth multiple bond breakings and bond formations. The geometrical details are given in the Appendix A, Section 4.

All calculations were performed using the Gaussian 16 software package [51] at the LEO HPC facilities of the University of Innsbruck.

## 5. Conclusions

In conclusion, we have comprehensively investigated the reduction of hexachlorobenzene and dichlorobenzene isomers in electron transfer processes and obtained selective C–Cl bond excisions for both molecular compounds in the gas-phase. This was achieved by the combined balance between the bond dissociation energies and electron affinity of the chlorine molecules. The reactions were all endothermic and, corresponding to the different isomers, the presence of a barrier resulted in an energy balance that favored the *ortho* configuration. The efficiency in producing Cl_2_^−^ was not particularly significant from the TOF negative ions branching ratios relative to other fragment anions; this was due to the relevant energy redistribution within the TNI upon electron capture that involved the different internal degrees of freedom through relevant vibronic coupling. The present findings may have implications regarding the role of a transition state in the chemical reactions which are prevalent in different environments, i.e., not only in the gas but in bulk systems. The sensitization of *ortho* positions in C_6_Cl_6_ and C_6_H_4_Cl_2_ could be achieved upon molecular reduction, which, in the present case, was achieved using a potassium atom, making these geometrical effects in polyatomic molecules important in single electron transfer reactions. We anticipate that within the scope of redox reactions, the effect of an electron donor (oxidized species) and an electron acceptor (reduced species) may help in determining the role of reactants and products.

## Figures and Tables

**Figure 1 molecules-27-04820-f001:**
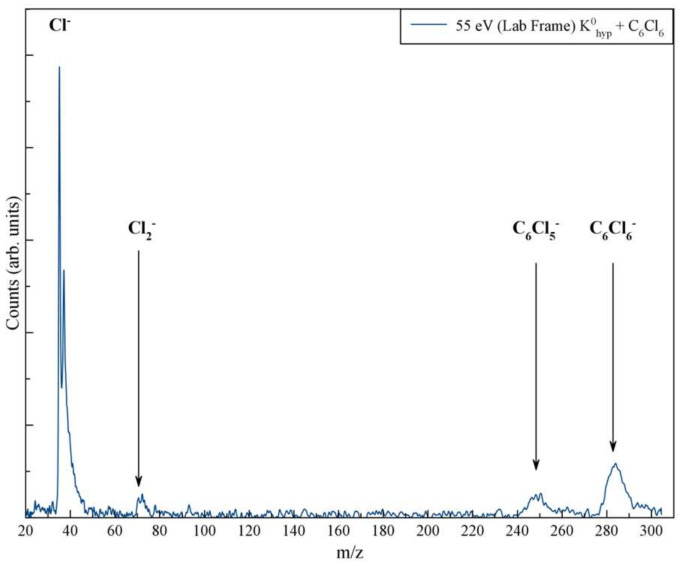
Time-of-flight negative ion mass spectrum in K–C_6_Cl_6_ collisions at 55 eV lab frame energy (43.5 eV in the center-of-mass frame). The spectrum shows the intensity of m/z anions.

**Figure 2 molecules-27-04820-f002:**
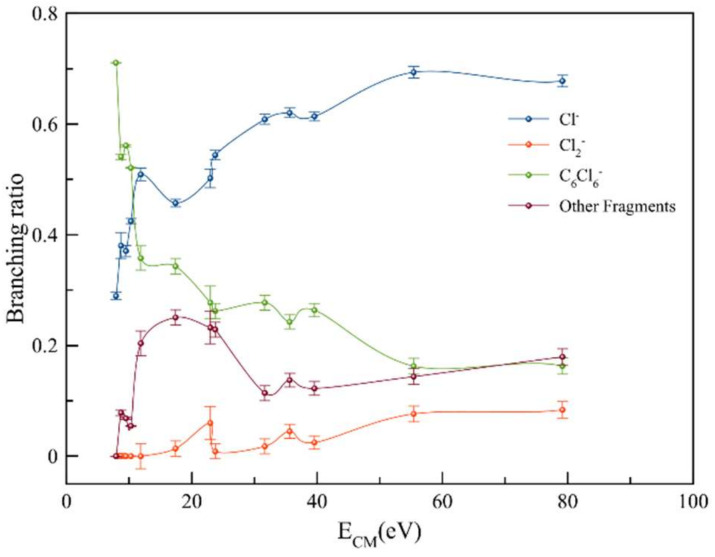
Hexachlorobenzene BRs (fragment anion yield/total anion yield) of C_6_Cl_6_^−^, Cl_2_^−^ and Cl^−^ ions formed as a function of the collision energy in the center-of-mass (CM) frame. Error bars are related to the experimental uncertainty associated with the ion yields. The lines serve purely to guide the eye.

**Figure 3 molecules-27-04820-f003:**
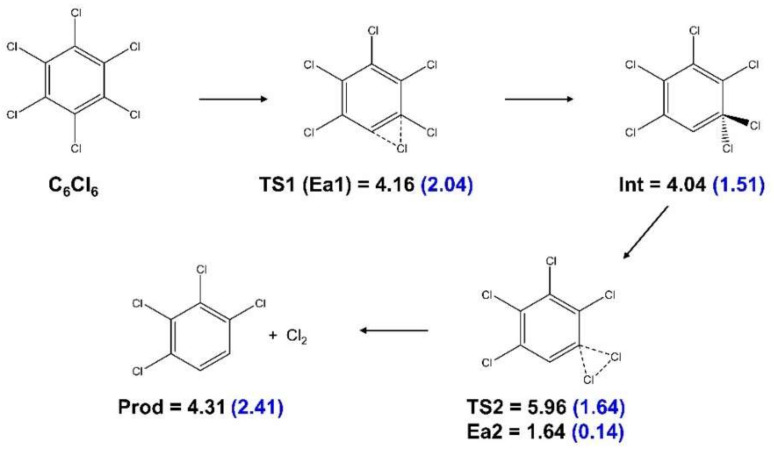
Minimal energy reaction pathway for the detachment of Cl_2_ and Cl_2_^−^ from *ortho* sites in C_6_Cl_6_. Energies in eV. The values in parenthesis are for the anionic system.

**Figure 4 molecules-27-04820-f004:**
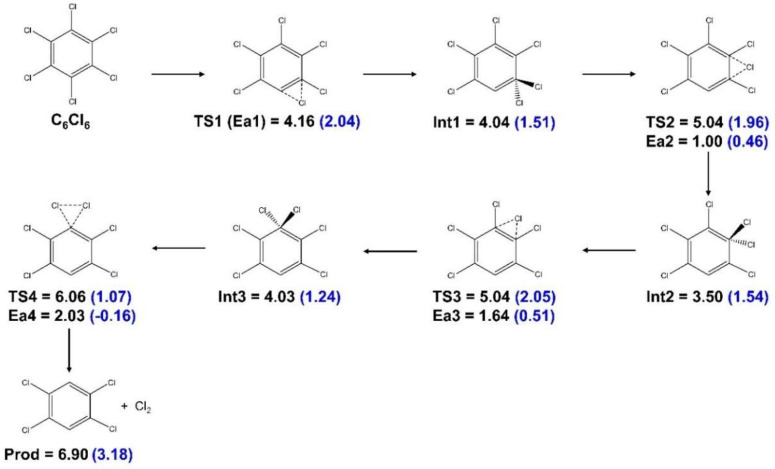
Detachment of Cl_2_ and Cl_2_^−^ from *para* sites in C_6_Cl_6_. Energies in eV. The values in parenthesis are for the anionic system.

**Figure 5 molecules-27-04820-f005:**
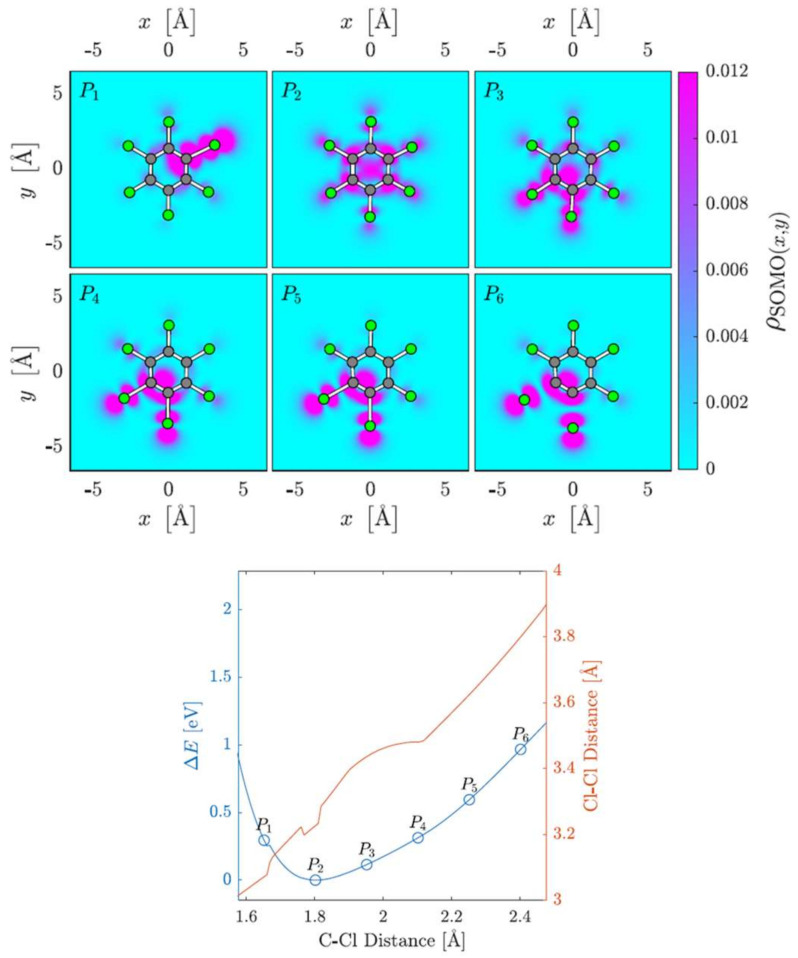
SOMO densities at six different C–Cl internuclear stretching distances and the corresponding potential energy curve.

**Figure 6 molecules-27-04820-f006:**
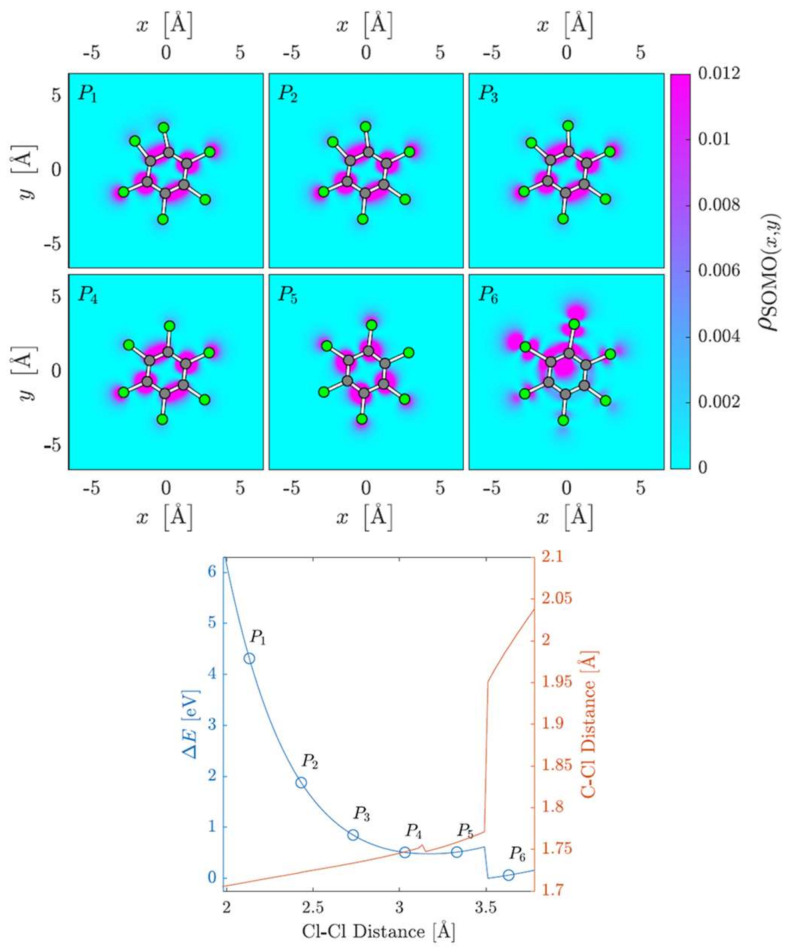
SOMO densities at six different C–Cl internuclear bending distances and the corresponding potential energy curve.

**Table 1 molecules-27-04820-t001:** Calculated Cl_2_^−^ appearance energies (AE), bond dissociation energies (D) and electron affinity (EA) for hexachlorobenzene at different levels of theory. Values are in (eV).

	CBS-QB3 (eV)	G4MP2 (eV)
AE(Cl_2_^−^),*o*-C_6_Cl_6_	1.97	1.96
AE(Cl_2_^−^),*m*-C_6_Cl_6_	2.39	2.39
AE(Cl_2_^−^),*p*-C_6_Cl_6_	3.66	3.65
*D*(C_6_Cl_5_-Cl)	4.38	4.18
*D*(*o*-C_6_Cl_4_-Cl)	2.66	2.64
*D*(*m*-C_6_Cl_4_-Cl)	3.07	3.08
*D*(*p*-C_6_Cl_4_-Cl)	4.35	4.34
*D*(Cl–Cl)	2.57	2.51
*EA*(Cl_2_)	2.50	2.36

**Table 2 molecules-27-04820-t002:** Calculated Cl_2_^−^ appearance energies (AE), bond dissociation energies (D) and electron affinity (EA) for hexachlorobenzene at different levels of theory. Values are in (eV).

Title 1	CBS-QB3 (eV)	G4MP2 (eV)
AE(Cl_2_^−^),*o*-C_6_H_4_Cl_2_	2.08	2.07
AE(Cl_2_^−^),*m*-C_6_H_4_Cl_2_	2.78	2.79
AE(Cl_2_^−^),*p*-C_6_H_4_Cl_2_	3.29	3.29
*D*(*o*-C_6_H_4_Cl-Cl)	4.41	4.23
*D*(*m*-C_6_H_4_Cl-Cl)	4.38	4.22
*D*(*p*-C_6_H_4_Cl-Cl)	4.41	4.25
*D*(*o*-C_6_H_4_-Cl)	2.74	2.71
*D*(*m*-C_6_H_4_-Cl)	3.46	3.44
*D*(*p*-C_6_H_4_-Cl)	3.95	3.91

## Data Availability

Not applicable.

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
