# Peer review of "Sensing the ortho Positions in C6Cl6 and C6H4Cl2 from Cl2 Formation upon Molecular Reduction"

_molecules, 2022, doi:10.3390/molecules27154820_

Round 1

Reviewer 1 Report

In this manuscript, the reduction of hexachlorobenzene in the electron transfer reaction was studied by combining experiments with theoretical calculations, and the influence mechanism of the position of chlorine atoms around the aromatic ring was discussed. It was found that Cl2- could only be formed in 1, 2-C6H4Cl2, the two nearest C-Cl bonds in the molecule were cleaved, and the chlorine atoms gathered together in the ring framework. It is found that the strong coupling between electrons and C-Cl bending motion leads to the position isomerization effect. It is a very interesting work. The manuscript is well done, and the data is full and accurate. The following suggestions are for the author's reference and can help the authors improve the quality of the manuscript.

1. The software G16 used by in present manuscript should be cited.

2. How does the author confirm that TS1, TS2 and intermediates of the reaction pathway in the Supplemental Material Files are correct, and whether the optimization is restricted optimization or full optimization, these should be explained in the manuscript? It is suggested that the author should make a comparative analysis and confirmation in combination with the reaction (2) and (3) in the Section 2.

3. Whether the basis sets and methods used by the author are suitable for the system studied should also be explained.

Author Response

Please see attached file with responses to reviewer #1

Reviewer 2 Report

The authors reported a combined study of experimental and theoretical analysis of the hexachlorobenzene and dichlorobenzene reduction processes. The experimental technique is the well established cross atom-molecular beam, while the electronic structure calculations employed the density functional theory with the B3LYP-GD3 functional and AVTZ basis set. Other theoretical methods such as G4MP2 and CBS-Q3 have been employed.

Comments:

1. The structure of the manuscript does not help the readability. I would request the authors to change section 4 to the position before the Results. It is difficult to rationalize the results and discussion content without knowing the experimental techniques and the theoretical calculations.

2. It is missing in the introduction what is the main objective of the work and what are the methods employed.

3. Fig S1 and S2 should be included in the text, since it will help on the vizualization of the mechanism involved 

4. The computational methods section does not have a single reference for the code, methods and basis sets. The authors should include the proper references.

5. Line 112: If a previous study used the wB97XD functional, what was the reason to use B3LYP in the current case? If previous results show good agreement, I see no reason to arbitrarily change the functional.

6. Line 132: Why also mixing the results of B3LYP with M06-2X and family type of basis set from Poople to Dunning?  

7. Many "see" Supplementary Information is found along the manuscript. For the sake of readability, the authors could include relevant information in the manuscript and/or make a reference to which Figure of Table from the SM that they are describing. 

8. The authors refer in the conclusions to a barrier existence in the reactive process, while in Line 216 they send to SM. Have these barriers been characterized from the theoretical perspective? Figure S4 reports an one dimension cut on the multidimensional potential energy surface. It is not clear how this cut is obtained, although the SM makes a reference to see text. 

9. Table S1 shows thresholds for the dissociation, how these values have been obtained? Separated fragments? Stretched optimized bonds? 

10. With such a large value of available energy, I would expect that excited electronic states could be of relevance in the process. I would improve the article if a comment on the subject appears on the manuscript. 

11. Some typos may be found along the text. For example, in line 235 “that did not charge exchange”, Line 262, missing “respectively”.

Overall, I think that the manuscript could be accepted if the questions/comments are taken into account.

Author Response

Please see attached file with responses to reviewer #2

Round 2

Reviewer 2 Report

The authors have answered all my questions and commented clearly on my main concerns related to the manuscript.